# Early warning for healthcare acquired infections in neonatal care units in a low-resource setting using routinely collected hospital data: The experience from Haiti, 2014–2018

Annick Lenglet[1,2]*, Omar Contigiani[1,3,4], Cono Ariti[5], Estivern Evens[6], Kessianne Charles[6], Carl-Frédéric Casimir[6], Rodnie Senat Delva[6], Colette Badjo[6], Harriet Roggeveen[1], Barbara Pawulska[1], Kate Clezy[1], Melissa McRae[1], Heiman Wertheim[2], Joost Hopman[1,2,7]

1 Médecins Sans Frontières, Amsterdam, The Netherlands, 2 Department of Medical Microbiology, Radboud Institute for Health Sciences, Radboudumc, Nijmegen, The Netherlands, 3 Leiden Observatory, Leiden University, Leiden, The Netherlands, 4 Lorentz Institute for Theoretical Physics, Leiden University, Leiden, The Netherlands, 5 School of Medicine, Cardiff University, Cardiff, United Kingdom, 6 Médecins Sans Frontières, Port au Prince, Haiti, 7 Department of Patient Safety and Quality, Radboudumc, Nijmegen, The Netherlands

* annick.lenglet@amsterdam.msf.org

**Data Availability Statement:** MSF has a managed access system for data sharing that respects MSF's legal and ethical obligations to its patients to

## Abstract

In low-resource settings, detection of healthcare-acquired outbreaks in neonatal units relies on astute clinical staff to observe unusual morbidity or mortality from sepsis as microbiological diagnostics are often absent. We aimed to generate reliable (and automated) early warnings for potential clusters of neonatal late onset sepsis using retrospective data that could signal the start of an outbreak in an NCU in Port au Prince, Haiti, using routinely collected data on neonatal admissions. We constructed smoothed time series for late onset sepsis cases, late onset sepsis rates, neonatal care unit (NCU) mortality, maternal admissions, neonatal admissions and neonatal antibiotic consumption. An outbreak was defined as a statistical increase in any of these time series indicators. We created three outbreak alarm classes: 1) thresholds: weeks in which the late onset sepsis cases exceeded four, the late onset sepsis rates exceeded 10% of total NCU admissions and the NCU mortality exceeded 15%; 2) differential: late onset sepsis rates and NCU mortality were double the previous week; and 3) aberration: using the improved Farrington model for late onset sepsis rates and NCU mortality. We validated pairs of alarms by calculating the sensitivity and specificity of the weeks in which each alarm was launched and comparing each alarm to the weeks in which a single GNB positive blood culture was reported from a neonate. The threshold and aberration alarms were the strongest predictors for current and future NCU mortality and current LOS rates (p<0.0002). The aberration alarms were also those with the highest sensitivity, specificity, negative predictive value, and positive predictive value. Without microbiological diagnostics in NCUs in low-resource settings, applying these simple algorithms to routinely collected data show great potential to facilitate early warning for

collect, manage and protect their data
responsibility. Ethical risks include, but are not
limited to, the nature of MSF operations and target
populations being such that data collected are often
highly sensitive. Data are available on request in
accordance with MSF's data sharing policy
(available at: http://fieldresearch.msf.org/msf/
handle/10144/306501). Requests for access to
data should be made to data.sharing@msf.org.

**Funding:** The authors received no specific funding
for this work.

**Competing interests:** AL, EE, KC, CFC, RSD, CB,
HR, BP, KC, and MM are employees of Medecins
Sans Frontieres. CA works as a statistician
consultant for MSF and consults on analysis plans
for all research activities for this Operational Centre
of MSF. HW and JH are employees of
Radboudumc. There are no patents, products in
development or marketed products to declare. This
does not alter our adherence to PLOS ONE policies
on sharing data and materials.

possible healthcare-acquired outbreaks of LOS in neonates. The methods used in this study require validation across other low-resource settings.

## Introduction

Neonatal sepsis remains a significant worldwide cause of morbidity and mortality. It is currently estimated that 3 million cases of neonatal sepsis occur globally on an annual basis with mortality ranging between 11 and 19% [1]. These estimates are from studies exclusively based in high- and middle-income countries and predominantly hospital-based studies. They are therefore not reflective of the burden of neonatal sepsis in low-resource settings, nor accurately estimate the community-acquired rates of neonatal sepsis [1]. It is expected that the highest burden of neonatal sepsis is carried by low-and middle-income countries [2] and that this burden is higher for both community and healthcare-acquired (HA) sepsis [3].

HA infections (HAIs) are an important cause of sepsis in intensive care units. Global estimates attribute 110 out of 1000 admitted neonates suffering from HA sepsis, representing more than 50% of all HAIs in this setting [4]. HAI outbreaks in neonatal intensive care units (NICUs) represent 38% of all reported NICU outbreaks and 88% of all outbreaks reported in neonatology [5]. Most bacterial pathogens implicated in NICU outbreaks have been from Gram negative bacteria (GNB), sometimes multi-drug resistant (MDR)[5]. Prevention of outbreaks in NICUs is based on a comprehensive approach that includes appropriate staffing ratios, antibiotic stewardship, infection prevention and control (IPC) activities (appropriate spacing between beds, multi-modal hand hygiene measures, transmission based precautions, cleaning of equipment and environment etc.) and surveillance for suspected HAIs (both clinical recognition as well as microbiological diagnosis when possible) [6].

Late onset (LO) sepsis in neonates is defined as sepsis with an onset more than 48 hours after birth[7]. It is generally accepted to be HA. Early recognition of a potential outbreak in a neonatal ward can be through the recognition of an unusual increase of suspected LO sepsis or through the identification of bacterial isolates (from blood culture or cerebrospinal fluid from septic neonates) of the same species in samples from multiple patients. In low-resource settings, the early identification of potential NICU outbreaks is complicated by multiple factors: workload of clinical staff (reducing ability to recognise unusual increases in septic neonates or inability to report such observations), limited data management or data collection systems around hospital admissions and absence of microbiological testing capacity to confirm suspected HAI outbreaks. Furthermore, tackling hospital based outbreaks can be time and resource consuming, as multiple transmission routes (including: environmental contamination, colonized healthcare staff, contaminated medical devices and products) might be implicated and need to be investigated and addressed. The ability therefore in these contexts to have an early warning on possible NICU outbreaks through alternative surveillance indicators [8], could be invaluable as it could prevent morbidity and mortality.

The early detection of infectious disease outbreaks using automated syndromic surveillance systems is not novel. A review of 35 evaluations for outbreak detection using automated syndromic surveillance systems in 2007 showed that they can detect large seasonally occurring outbreaks with sensitivity and timelines that are comparable or better than systems that rely on diagnostic data [9]. The use of automated detection for HA outbreaks has also gained interest in high resource settings in recent years [10, 11]. Most of these initiatives are based on data mining of existing hospital electronic health records (EHRs), which include microbiological diagnostic date and increasingly through detection of clusters using Whole Genome

Sequencing (WGS). A university hospital in the United States, used historical data from EHRs and WGS from 2011 to 2016 (when nine HA outbreaks were detected), to show that 78% of HA infections could have been prevented as the automated data mining was able to identify routes of exposure associated with those outbreaks [12]. In Germany, automated screening and systematic analysis of routinely collected microbiological laboratory data and patient location, was able to detect six out of seven HA outbreaks from rare pathogens, whereas the results for more frequently identified pathogens were variable [13]. In four units from a 5000-bed hospital in Wuhan (China), they showed that using a combination of data on antibiotic utilization rates, the number of bacterial samples requested and sample positivity rate, they were able to detect HAI clusters, even though the accuracy of the detection was variable between hospital units [14]. In South Africa, the combination of laboratory and antimicrobial usage data had high sensitivity and positive predictive value for HAI determination in hospitalised paediatric patients [15]. Similar studies for hospitalised neonates are not available despite a recent call to increase research to identify surveillance systems that raise an alert when a pre-set trigger indicates an outbreak has occurred in these settings [16].

Haiti is the poorest country in the Western Hemisphere with an estimate 6 million Haitians living below the poverty line (USD 2.41 per day) and 2.5 million Haitians living under extreme poverty (USD 1.12 per day) [17]. Between 2011 and 2018, Médecins Sans Frontières (MSF) managed an emergency obstetric emergency hospital and its associated neonatal care unit (NCU; not relying on central or umbilical catheters) in Port-au-Prince, Haiti. Between July 2014 and October 2015, the NCU experienced an outbreak of infections of HA extended-spectrum β-lactamase (ESBL)-producing *Klebsiella pneumoniae*, which has been described previously [18][NO_PRINTED_FORM]. Until the closure of the hospital in July 2018, repeated clusters in the NCU associated with multi-drug resistant (MDR)-GNB continued to occur (MSF unpublished data). Recognition of these outbreaks was through an unusual increase in neonatal mortality and through clinical astuteness observing an unusual increase in neonates with LO sepsis. Also, microbiological diagnosis from blood cultures from these neonates conducted in an external private laboratory showed a cluster of bloodstream infections from the same bacterial species with similar antibiotic resistance profiles.

We aimed to explore whether we would have been able to generate reliable (and automated) early warning for potential clusters of LO sepsis that could signal the start of an outbreak in the NCU and thus reducing the reliance on astute clinicians or on a microbiological laboratory. Such early warning systems could be implemented across other humanitarian settings with NCUs as an early warning for suspected HAI outbreaks in admitted neonates.

## Methods

### Setting

CRUO (Centre de Réference des Urgences Médicales) was a container hospital (established soon after the earthquake in 2010) in Port Au Prince. It closed in 2018. It was equipped with 162 beds, including 106 obstetric beds and 56 for neonates in the NCU. The hospital ran on city power when available and was otherwise supplied with external generators. Water was supplied to the hospital by trucks from a private company. All water was filtered and chlorinated on site of the hospital to render it potable. The NCU provided phototherapy, incubators, and Continuous Positive Airway Pressure (CPAP) machines for neonates with breathing difficulties. After admission to the NCU following their birth at CRUO, neonates clinically judged to be at risk for sepsis were administered pre-emptive antibiotics. This was based on prematurity, low birth weight, chorioamnionitis, or other materno-fetal risk factors. The NCU did not use central or umbilical catheters which is why we do not refer to it as a NICU [18].

Microbiological diagnosis of blood cultures was available at a private laboratory in Port au Prince for payment. A case of neonatal sepsis was defined as a neonate in the neonatal unit who presented with one or more of the following clinical signs 48 hours after birth: prolonged capillary refill, certain skin changes (redness, sclerema), distended abdomen with/without hemorrhagic, brownish or bilious gastric aspirates, tachypnea, tachycardia, persistent jaundice, unstable temperature, signs of disseminated intravascular coagulation (bleeding from catheter sites, bloody secretions from nose and mouth, petechiae), reduced muscle tone, lethargy, apathy or irritability [18]. The case definition did not include any microbiological criteria due to the difficulty in accessing this diagnostic tool in this setting.

## Data sources

We used routinely collected data at the hospital which was available in five different (Excel-based) datasets: maternal admissions, neonatal admissions, sepsis line list for neonates, microbiological results from neonatal blood cultures and NCU pharmacy consumption data for first (ampicillin and gentamicin), second (ceftazidime and amikacin) and third line (meropenem) antibiotics for neonatal sepsis treatment. The study period included July 2014 until December 2017.

## Data analysis

**a) Analytical approach.**   We had no pre-existing definition for an outbreak of healthcare-acquired neonatal sepsis in CRUO, and microbiological data was only available for a small proportion of suspected neonatal sepsis patients. We therefore defined an outbreak in the NCU as a moment in time in which we observed statistical abnormalities in any of the routinely collected health indicators. We used data from all five data sources to construct time series and to study their statistical abnormalities as suspected outbreaks. To do this, from the time series we defined outbreak indicators and outbreak alarms. Outbreak indicators were represented by smoothed time series built from the retrospective data. Outbreak alarms were binary variables and were triggered when time series' abnormalities might indicate an outbreak. They represented simple measures, easy to obtain from routinely collected data in real-time on a weekly basis. The outbreak indicators described more robust measures, protected from week-by-week statistical fluctuations, and therefore could be used to validate outbreak alarms. This validation was done by assuming that outbreak indicator variables had a higher value during outbreaks compared to non-outbreak periods. Furthermore, because of their underlying connection, we expected the values of valid outbreak indicators to be highly correlated with each other. This mutual consistency between different indicators and alarms was used to validate the approach in the absence of a gold standard definition of outbreak.

**b) Outbreak indicators (smoothed time series).**   We constructed outbreak indicator time series by epidemiological week for the following indicators: number of neonatal admissions, maternal exits, maternal admissions with normal pregnancies, LOS cases (based on date of admission), LO sepsis rate (LO sepsis cases/total NCU admissions), overall mortality (total neonatal deaths/total neonatal exits), number of positive blood cultures for GNB and consumption of first, second- and third-line antibiotics. We also created a time series combining the second- and third-line antibiotic consumption.

We defined the corresponding outbreak indicator of each time series as a smoothed version of it, obtained by calculating centred moving averages in a five-week period (the mean value calculated over a five-week period). We chose a window of five weeks, because significant autocorrelations of LO sepsis rates were present only for time lags below 5 weeks. In addition, we constructed two displaced moving averages that represented the average of the LO sepsis

and mortality rates three weeks in the future ("future LO sepsis cases" and "future mortality"). We verified the existence of seasonality through visual inspection.

**c) Outbreak indicator correlations.** Under the assumption that all the outbreak indicators were associated with outbreak events, we first estimated the level of correlation between each combination of indicators. To do this, we repeatedly calculated the mean of each of these indicators over 25 randomly selected weeks to obtain a distribution of the means for this specific outbreak indicator. The choice of using 25 weeks was based on two factors: 1) mathematically, considering many weeks (>10) ensures that the mean values follow an approximately normal distribution, for which the correlation coefficient is well-defined; and 2) this number of weeks corresponds to the minimum number of weeks marked an outbreak alarm. We performed a sensitivity analysis by modifying the number of weeks for which the repeated means were calculated and verified that it did not affect our conclusions. From the means distributions we then computed the Pearson correlation coefficient for every possible combination of two outbreak indicators (i.e. LO sepsis rates with NCU mortality, LO sepsis rates with maternal admissions, LO sepsis rates with neonatal admissions etc.). We selected the indicators with the highest correlation coefficient (r>0.6), assuming that they were more likely to identify weeks in which an outbreak occurred. This cut-off was arbitrary and should be adjusted depending on the dataset considered.

**d) Outbreak alarms.** To define outbreak alarms, we chose indicators that displayed high correlation with weeks in which a likely outbreak occurred. We defined three classes of potential early warning alarms for outbreaks in the NCU: 'threshold alarms', 'differential alarms' and 'aberration alarms'. The choice for these three classes of alarms was because they would be easy to generate in any setting as they did not require complex data sets nor extensive data manipulation. The alarm classes and the indicators on which they were based are summarised in Table 1. Threshold alarms were fixed thresholds for the weekly number of LO sepsis cases, weekly proportion of LO sepsis (proportional morbidity) and overall neonatal mortality. The threshold for LO sepsis cases and proportional morbidity were chosen as these values would have stood out to clinical staff of CRUO as an anomaly. The threshold for mortality was chosen as the 'normal' limit for NCU mortality in CRUO which was 15%. Differential alarms were constructed for weekly numbers of LO sepsis cases and overall neonatal deaths and were triggered in the weeks that these indicators were 100% greater than the same values in the previous week. Aberration alarms were created for LO sepsis rate and overall neonatal mortality using the improved Farrington algorithm described by Höhle and Mazick [19] and based on methods described by Farrington et al. [20] and Noufaily et al. [21]. They were constructed by

**Table 1. Classes of outbreak alarms, how they are defined and their relevance.**

| Alarm class | Indicator | Definition | Implication |
|---|---|---|---|
| **Threshold** | *LO sepsis cases* | Weekly LOS cases exceed 4 cases | All indicators could be tracked by clinical staff easily on a daily and weekly basis |
| | *LO sepsis rate* | Weekly LOS rate exceeds 10% of total NCU admissions | |
| | *Mortality* | Weekly overall NCU mortality exceed 15% | |
| **Differential** | *LO sepsis cases* | Weekly LOS cases double compared to previous week | All indicators could be tracked by clinical staff easily on a daily and weekly basis |
| | *Mortality* | Weekly mortality double compared to previous week | |
| **Aberration** | *LO sepsis rate* | Improved Farrington algorithm applied to current and historic weekly LOS rates | Would require data analytical skills in the team to apply to weekly routine data |
| | *Mortality* | Improved Farrington algorithm applied to current and historic weekly NCU mortality | |

modelling the time series for LO sepsis rate and overall mortality through a generalised linear Poisson model. For every week in the time series, the model identified whether the values of LO sepsis proportional morbidity and overall mortality were significantly higher (p-value α<0.05) compared with the same weeks in other years and the weeks before and after (window half-width of 3 weeks).

**e) Validation and performance of outbreak alarms.**   For every combination of outbreak alarms and outbreak indicators, we first calculated the mean value of the indicator in the weeks marked by the alarm. We then estimated the significance of this mean value to assess the performance of each alarm. To calculate this, we obtained the distribution of the mean values of the indicator for a random alarm marking 'n' weeks, through Monte Carlo simulations, by resampling without replacement 'n' times the original indicator time series. Repeating this process produced a probability density function for the mean value and a p-value for the mean value produced by the real outbreak alarm.

To quantify the consistency between the different alarms, we calculated four performance metrics for every pair of alarms. These are sensitivity, specificity, positive predictive value (PVV) and negative predictive value (NPV). We also tested the performance of our outbreak alarms against microbiological data. We constructed a validation alarm which was triggered each week in which a single GNB positive blood culture was reported. Using the metrics defined above, we compared the threshold, differential, and aberration alarms against the GNB positive blood culture alarm. In this validation process, we considered that a week in which this latter alarm was triggered was a week in which there was an LO sepsis outbreak present. Finally, to further quantify the agreement between the calculated alarms, we performed a McNemar's test to test the concordance between weeks in which an alarm was raised for pairs of alarms.

All data management and analyses were conducted using R (Version 4.0.2), Rstudio (Version 1.3.1056) and Python (Version 3.5). Alarm signals were constructed using the R package "surveillance" version 1.19.1[22]. The comprehensive annotated coding used for the analysis is available in the following location: https://github.com/R4EPIarchive/HTI_OutbreakNCU_2021_August_Timeseries_HAI.

## Ethical considerations

This study was approved by the National Bioethical Committee of the Ministry of Public Health and Population of Haiti. This research fulfilled the exemption criteria set by the Médecins Sans Frontières Ethics Review Board for a posteriori analyses of routinely collected clinical data and thus did not require MSF ERB review. It was conducted with permission from Melissa McRae, Medical Director, Operational Centre Amsterdam (OCA), Médecins Sans Frontières.

Patient records included in this retrospective analysis covered the period in which they received treatment from January 2014 to December 2017. They were accessed in February 2022 to start the analysis for this study. Data were aggregated from existing individual health information system records and these aggregated datasets were used for the analysis of this study. The data collected was not collected specifically for the purpose of this study, but was part of routinely collected health information systems to monitor programme implementation. Patients therefore did not provide informed consent for the data to be collected.

## Results

### General

Time series for neonatal admissions, maternal exits, maternal admissions with normal pregnancies, NCU mortality and antibiotic consumption were available from January 2014 until

December 2017. However, data on LO sepsis admissions and positive blood cultures was only available from July 2014 as that is when the first outbreak was observed, and additional data collection started. We therefore restricted the analysis for the remainder of the study to data from July 2014 to December 2017 (182 consecutive weeks of data). Visual inspection suggested a strong seasonality in the time series for neonatal and maternal admissions with peaks corresponding to the period between October and December each year (Fig 1). No seasonality was visible in the time series of the other indicators.

## Correlation of indicators and outbreak definition

We identified two distinct groups of indicators with strong correlations (Pearson correlation coefficient > 0.6) (correlation matrix-Table 2). The first group were those indicators that are related to current and future rate of LO sepsis and mortality amongst admitted neonates in the NCU. The second group included those indicators that are related to the level of occupancy in the hospital in terms of total maternal exits, maternal admissions for normal pregnancies and neonatal admissions (Table 2). Because any causal link between indicators would result in a strong correlation, the lack of correlation between indicators in these two separate groups implies that a causal relationship was unlikely. The correlations between antibiotic consumption and GNB positive blood cultures and all the other indicators were also very weak which suggests that they cannot be used to identify suspected LO sepsis outbreaks in the NCU. Based on these results, we concluded that LO sepsis, admissions and NCU mortality (both present and future time series) were the strongest indicators for suspected LO sepsis outbreaks in the NCU. The alarm classes were therefore tested against current and future LO sepsis rate and NCU mortality.

## Outbreak alarms

We identified 7 outbreak alarms (threshold = 3, differential = 2, aberration = 2) and measured them against four possible outbreak indicators (current LO sepsis rate, current NCU mortality, future LO sepsis rate and future NCU mortality). As an example, Figs 2 and 3 show the threshold alarm for LO sepsis rates and how often it is triggered in the outbreak indicators (smoothed time series) for LO sepsis weekly rate and NCU mortality (S1 File: Supporting information includes the other outbreak alarms and indicators). We do not include the alarm constructed for the GNB positive blood cultures in this as it is used only for validation purposes.

For the study period, the LO sepsis case threshold alarm was triggered the greatest number of weeks (n = 47), followed by the LO sepsis rate threshold (n = 37) and the mortality aberration alarm (n = 31) (Table 3). The LO sepsis rate aberration and mortality threshold alarm were triggered 29 weeks, the differential mortality alarm for 27 weeks and the LO sepsis rate differential alarm 25 weeks. Threshold and aberration alarms were strong predictors for both current and future NCU mortality and LO sepsis rate (p-value<0.002; Table 3). They also predict very well weeks in which a GNB positive blood culture was reported (p-value<0.002). In comparison, the differential alarms performed poorly for LO sepsis admissions (p>0.02). They performed better in predicting future mortality and weeks with GNB positive blood cultures (p-value = 0.02).

## Validation of alarms

LO sepsis alarms displayed the highest sensitivity (range 38–72%) and specificity (>86%) with other alarms. Similarly, the mortality alarms had the highest sensitivity and specificity with other alarms based on the same indicator (Table 4). The alarm with the highest sensitivity and

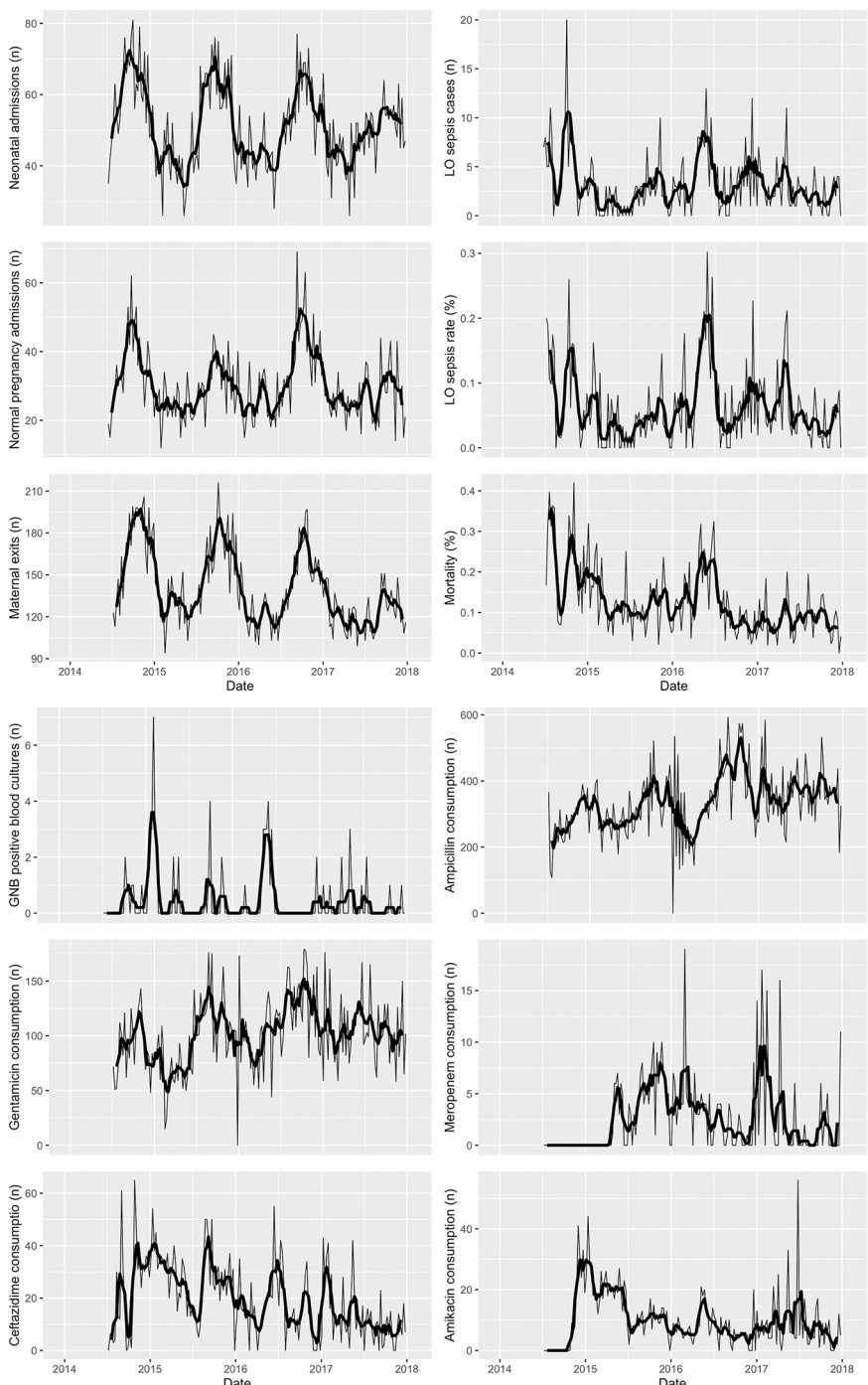

**Fig 1.** A and B: Weekly time series with five weekly moving average (thick black line) for routinely collected data on neonatal admissions, maternal exits, maternal admissions for normal pregnancies, neonatal mortality, LO sepsis cases, LO sepsis rates, GNB positive blood cultures and antibiotic consumption, July 2014 to December 2017, CRUO.

specificity for weeks in which the GNB positive blood culture alarm was triggered was the LO sepsis rate aberration alarm (Table 4). In general, the specificity for all alarms was high (>84%) with weeks in which the GNB positive blood culture alarms were triggered. The sensitivity of all the alarms remained below 50% for the weeks in which the GNB positive blood

**Table 2. Correlation matrix displaying the Pearson correlation coefficient between different outbreak indicators.**

| | LO sepsis rate | Mortality | Future Mortality | Future LO sepsis rate | GNB positive BCs | 2nd and 3rd line Abs | Maternal exits | Normal pregnancy admissions | Neonatal admissions |
|---|---|---|---|---|---|---|---|---|---|
| **LO sepsis rate** | | 0.6 | 0.6 | 0.8 | 0.5 | -0.06 | -0.06 | -0.005 | -.01 |
| **Mortality** | | | 0.8 | 0.4 | 0.4 | 0.1 | 0.1 | -0.03 | 0.009 |
| **Future Mortality** | | | | 0.6 | 0.3 | 0.03 | 0.1 | -0.02 | -0.01 |
| **Future LO sepsis rate** | | | | | 0.4 | -0.2 | -0.002 | 0.07 | -0.1 |
| **GNB positive BCs** | | | | | | 0.3 | -0.05 | -0.07 | -0.1 |
| **2nd and 3rd line Abs** | | | | | | | 0.2 | -0.08 | -0.007 |
| **Maternal exits** | | | | | | | | 0.8 | 0.9 |
| **Normal pregnancy admissions** | | | | | | | | | 0.9 |

culture alarms were triggered except for the LO sepsis rate aberration alarm (sensitivity = 66%; Table 4). These results are reenforced from the positive and negative predictive values for the different alarm pairs (Table 5). Using McNemar's test, we find that the LO sepsis threshold alarm is significantly different from the others (Table 6). This is due to the higher number of weeks that this alarm was triggered and confirms that case threshold is a simple, albeit conservative choice.

## Discussion

We have shown that both threshold and aberration alarms for LO sepsis cases, LO sepsis rates and overall neonatal mortality were strong predictor alarms for weeks in which the reported LO sepsis rates and mortality were unusually high (and suspected to be weeks in which an outbreak was occurring in the neonatal unit). Using a threshold for LO sepsis cases was also shown to trigger more alarms than any other alarm category, rendering it a highly conservative

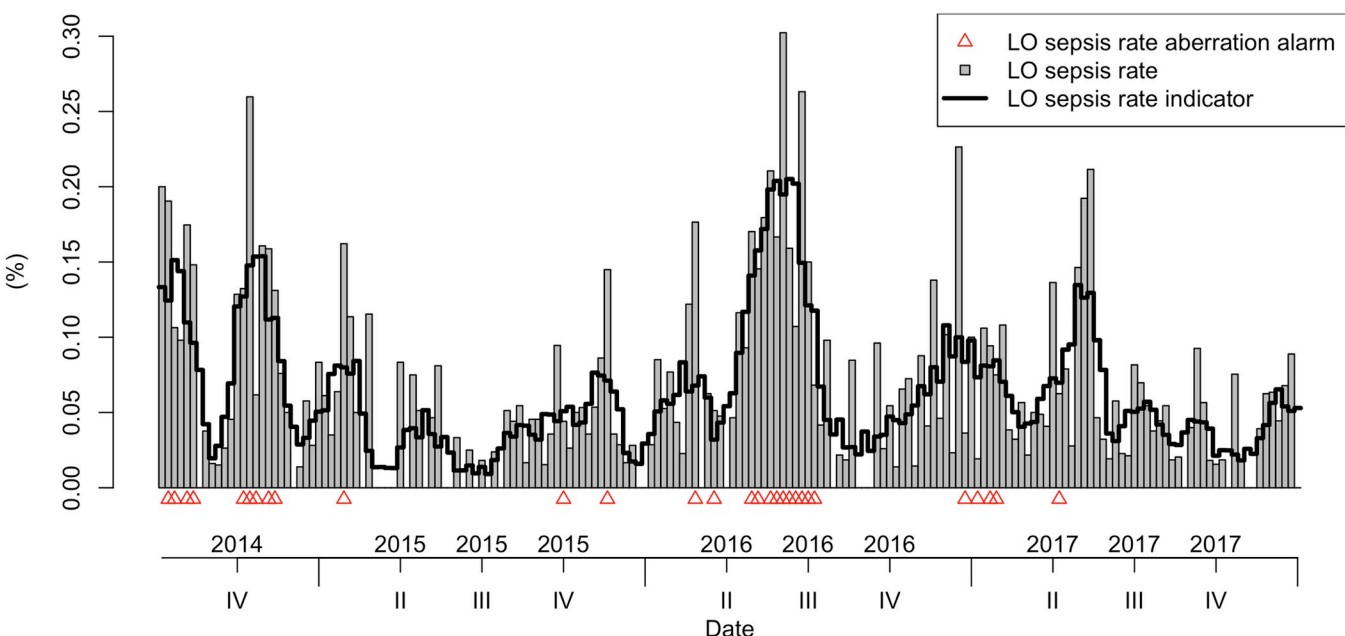

**Fig 2. LO sepsis rate threshold outbreak alarm (red triangles) and the current smoothed LO sepsis weekly rate.**

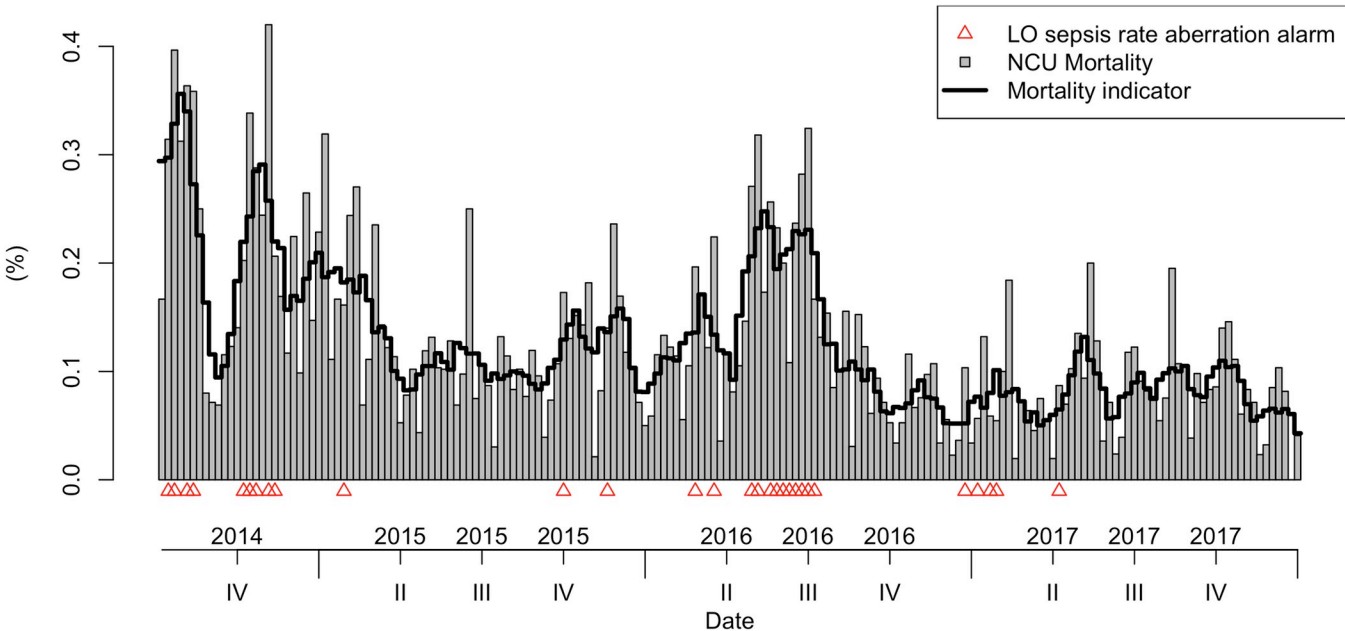

**Fig 3. LO sepsis rate threshold outbreak alarm (red triangles) and the current smoothed NCU mortality.**

alarm for suspected outbreaks. In contrast, differential alarms were not useful suspected outbreak predictors. We hypothesize that this may be due to high variability in daily LO sepsis cases and deaths and you would need to see a sustained change over a longer period for it to be useful, albeit not very timely. Another important finding is that increased mortality (through threshold or aberration detection) was also a strong predictor of increased mortality in the future three weeks, whereas this does not happen for alarms associated with LO sepsis cases. We are confident that strong predicting alarm classes are 'true' occurrences because they are clustered in time (which also suggests the occurrence of an outbreak) and because of the high negative predictive value compared to GNB positive blood cultures. These results suggest that simple threshold alarms can serve as early warnings for suspected neonatal outbreaks and potentially prevent additional morbidity and mortality if control measures are immediately taken.

**Table 3. Performance (p-values) of the alarm classes in predicting abnormal values of outbreak indicators and weeks in which GNB positive blood cultures are reported (columns).**

| Alarm classes | | Weeks alarm is triggered | Outbreak indicators and GNB positive blood culture weeks | | | | |
|---|---|---|---|---|---|---|---|
| | | | LO sepsis rate | NCU mortality | Future LO sepsis rate | Future NCU mortality | GNB positive blood culture weeks |
| Threshold | LO sepsis cases | 47 | <0.001 | <0.001 | <0.001 | <0.001 | <0.001 |
| | LO sepsis rate | 37 | <0.001 | <0.001 | <0.001 | <0.001 | <0.001 |
| | Mortality | 29 | <0.001 | <0.001 | <0.001 | <0.001 | <0.001 |
| Differential | LO sepsis cases | 25 | 0.07 | 0.7 | 0.3 | 0.4 | 0.6 |
| | Mortality | 27 | 0.2 | 0.3 | 0.02 | 0.2 | 0.02 |
| Aberration | LO sepsis rate | 29 | <0.001 | <0.001 | <0.001 | <0.001 | <0.001 |
| | Mortality | 31 | <0.001 | <0.001 | <0.001 | <0.001 | 0.001 |

[Note: p-values obtained through bootstrap process].

**Table 4. Sensitivity (Sens.) and specificity (Spec.) of pairs of alarm indicators.**

| | LO sepsis threshold | | LO sepsis rate threshold | | LO sepsis rate aberration | | Mortality threshold | | Mortality aberration | | GNB positive blood cultures | |
|---|---|---|---|---|---|---|---|---|---|---|---|---|
| | Sens. | Spec. | Sens. | Spec. | Sens. | Spec. | Sens. | Spec. | Sens. | Spec. | Sens. | Spec. |
| LO sepsis threshold | | | 72 | 98 | 45 | 94 | 38 | 92 | 43 | 92 | 40 | 86 |
| LO sepsis rate threshold | 92 | 91 | | | 54 | 94 | 46 | 92 | 49 | 91 | 46 | 86 |
| LO sepsis rate aberration | 72 | 83 | 69 | 89 | | | 59 | 92 | 62 | 92 | 66 | 88 |
| Mortality threshold | 62 | 81 | 59 | 87 | 59 | 92 | | | 93 | 97 | 48 | 84 |
| Mortality aberration | 65 | 82 | 58 | 88 | 58 | 93 | 87 | 99 | | | 48 | 85 |
| GNB positive blood cultures | 50 | 81 | 45 | 86 | 50 | 93 | 37 | 90 | 39 | 89 | | |

A recent review of 39 outbreaks in NICUs (35 in high income countries) determined that 33 had been detected through clinical case detection and microbiological confirmation or laboratory-based surveillance methods [23]. Furthermore, it is estimated that each neonatal care unit will experience up to 10 outbreaks per year [24]. Such outbreaks are also not always being detected in well-resourced hospitals [25]. Our findings show that threshold alarms for LO sepsis cases and neonatal mortality performed well to predict HAI outbreaks in an NCU (p<0.0001). This is therefore of particular importance in low-resource settings where microbiological screening or diagnosis of admitted neonates might not be available to detect or confirm outbreaks. Not only are these alarms simple indicators to collect, but they are also intuitive to apply and can be easily generated.

In our study, time series based on antibiotic consumption did not predict LO sepsis outbreaks or high mortality. We assumed that the consumption data around second and third line antibiotics would be particularly useful as these would only be used for treatment in neonates who showed signs and symptoms of neonatal sepsis or those that had indications of failing first-line treatment. Real time monitoring of antibiotic consumption was shown to be a useful proxy for healthcare associated infections in hospitalised children in South Africa [15]. It has also shown to be useful in predicting healthcare-acquired outbreaks in adult patient wards in China [14, 26]. We think that the lack of prediction in our study was likely due to antibiotic consumption data not being recorded in real time and for each individual neonate, thus not providing a 'live' picture of the clinical management of the neonatal patient population Consumption data in CRUO, was based on the weekly drug requests from the NCU to the central pharmacy and not on individual patient antibiotic consumption from an electronic medical record. It is also possible that because a high proportion of admitted neonates received antibiotics prophylactically, that antibiotic consumption was also not signalling a suspected outbreak.

**Table 5. Positive predictive value and negative predictive value for pairs of alarm indicators.**

| | LO sepsis threshold | | LO sepsis rate threshold | | LO sepsis rate aberration | | Mortality threshold | | Mortality aberration | | GNB positive blood cultures | |
|---|---|---|---|---|---|---|---|---|---|---|---|---|
| | PPV | NPV | PPV | NPV | PPV | NPV | PPV | NPV | PPV | NPV | PPV | NPV |
| LO sepsis threshold | | | 92 | 91 | 72 | 83 | 62 | 81 | 65 | 82 | 50 | 81 |
| LO sepsis rate threshold | 72 | 98 | | | 69 | 89 | 59 | 87 | 58 | 88 | 45 | 86 |
| LO sepsis rate aberration | 45 | 94 | 54 | 94 | | | 59 | 92 | 58 | 93 | 50 | 93 |
| Mortality threshold | 38 | 92 | 46 | 92 | 59 | 92 | | | 87 | 99 | 37 | 90 |
| Mortality aberration | 43 | 92 | 49 | 91 | 62 | 92 | 93 | 97 | | | 39 | 89 |
| GNB positive blood cultures | 40 | 86 | 46 | 86 | 66 | 88 | 48 | 84 | 48 | 85 | | |

**Table 6. Concordance between alarms indicators using McNemar's test.**

|  | LO sepsis threshold | LO sepsis rate threshold | LO sepsis rate aberration | Mortality threshold | Mortality aberration | GNB positive blood cultures |
|---|---|---|---|---|---|---|
| LO sepsis threshold | | 0.012 | **0.002** | **0.0044** | **0.0094** | 0.19 |
| LO sepsis rate threshold | | | 0.12 | 0.16 | 0.29 | 0.88 |
| LO sepsis rate aberration | | | | 1 | 0.68 | 0.095 |
| Mortality threshold | | | | | 0.41 | 0.15 |
| Mortality aberration | | | | | | 0.26 |
| GNB positive blood cultures | | | | | | |

A significant result (p-value<0.01, in bold) indicates that the two alarms are not concordant.

The two most common methods to validate hospital outbreak detection systems are either through simulations, in which the properties of an outbreak (e.g. duration, time of commencement) are fully specified beforehand, or through expert opinion, in which the outbreaks are manually identified using a complete dataset [11, 27]. A third method, sometimes called "derived" approach, uses outbreaks labels constructed from the data. However, because the specific criteria implemented in this process are usually informed by expert knowledge, this approach is commonly regarded as not being completely independent from the second approach. In this study we did not use a derived method to validate the identified outbreak alarms, so we did not rely on pre-defined outbreak and non-outbreak weeks. Instead, we quantified the similarities between different outbreaks definitions using the relative specificity and sensitivity. The strong correlation between our outbreak indicators time series might suggest that a binary classification for an outbreak was possible with the current dataset. However, we view the value of an alarm system not as its ability to recover some arbitrary outbreak label, but in its ability to predict clinical data which suggest the presence of an outbreak.

We accounted for seasonality in three ways in this study. Firstly, the most obvious visually identified trend was in the seasonality linked to maternal and neonatal admissions. This is a well described seasonal occurrence in Haiti as it falls nine months after Carnival [28]. Neither of these two indicators were our preferred outbreak indicators, so we did not consider their seasonality to impact on our outbreak indicators (LO sepsis rates and mortality). Secondly, we always considered LO sepsis rates and mortality rates where possible, thus accounting for the increases in maternal and neonatal admissions. Thirdly, the constructed aberration alarms already account for any seasonal trends in other years of the time series. In particular, the overlap between the aberration alarms and the simpler analogues of these alarms (threshold LO sepsis rate and threshold mortality) suggest that seasonality does not affect our outbreak classification and validation procedure. Furthermore, as our aim was to identify hospital outbreaks, by assuming they follow a seasonal pattern, we might discount them as an early warning signal of an actual outbreak.

The performance of the improved Farrington algorithm for the construction of aberration alarms has been debated [29, 30], with conflicting results depending on the size of the dataset used. Because the aberration alarms that we studied produced results that are consistent with other alarm classes, we find that this method performs adequately. It should be noted, however, that a full comparison depends on the specific setting, the average duration of outbreaks and the chosen significance threshold (e.g. $\alpha < 0.01$ instead of $\alpha < 0.05$). Due to the limited size of our dataset, we are not able to judge this accurately or quantify the performance of a more stringent significance threshold.

One of the main limitations in this study is that we were only able to use a single dataset to test our methods and findings (the predictor dataset also served as the validation dataset). As the data used to construct the alarms and test the alarms was generated by the same healthcare providers, a surveillance bias in reporting on LO sepsis cases or mortality might have been introduced (i.e. if they observed higher mortality, they started reporting more LO sepsis cases as well). This could have resulted in an overestimation of the strength of correlation between the different time series and strength of prediction of alarms with outbreak indicators. A second limitation is connected to the lack of complexity in the alarms and indicators we considered. The number of variables present in the raw data might imply that multivariate regression methods could be more successful. However, the length of the dataset and its large variance due to presence of many outbreaks did not allow us to exploit such techniques. Furthermore, the presence of low correlations between different classes of indicators did not allow for a more complex analysis. Similarly, the simple smoothing of the dataset using moving averages is also fully justified. Because the medical data used in this paper is collated weekly, we are free to use moving averages based on the length of an outbreak as opposed to using cumulative sum methods to account for reporting delays. A final limitation of our methodological approach that the threshold alarms were arbitrary and established using a context-informed approach. This approach meant that we understood the maximum new numbers of admissions to the NCU, we understood what mortality levels were acceptable in this very specific group of patients and had learnt to understand when the number of LO sepsis cases truly exceeded a normal situation. In addition, we understood how this data was being recorded and analysed. For this reason, the threshold alarms used in this study are likely not generalizable to other neonatal units and would require in depth contextual and epidemiological understanding before being defined.

In the context of low-resource settings, outbreaks in neonatal care units are frequent even though this burden remains poorly quantified [8] and microbiological diagnostics are limited [31]. As these units provide essential healthcare services to often already under-served communities, closing the units when suspected outbreaks from multi-drug organisms occur is not an option. Mitigation of suspected clusters of transmission or outbreaks is therefore the only option. The use of routine health indicators such as a threshold of LO sepsis cases or neonatal mortality are promising early warning signals for a suspected unusual event. Such alarms can easily be recognised and used to rapidly reinforce IPC measures (hand hygiene adherence, bed spacing, cleaning and disinfection and contact precautions) amongst healthcare staff and strengthen clinical management and microbiological diagnostics of any apparent LO sepsis cases to reduce further transmission of potentially pathogenic bacteria between vulnerable neonates. Early warnings for outbreaks therefore not only rely on astute clinicians, but in the absence of microbiological diagnostics, can be generated from simple data. The methods used in this study are simple to use and automate contexts with limited resources, lack of microbiology and minimal health information data. Clearly, this is a small study with a restricted dataset, thus future research will be required to validate whether our alarm signals will work in other neonatal care units (or whether some adjustment on the thresholds will be required). However, the methods applied have strong potential in the fight against HAI outbreaks in neonatal care units.

## Supporting information

**S1 File. Outbreaks alarms (red triangles) and indicators for LO sepsis.**
(DOCX)

## Acknowledgments

We would like to thank Kostas Danis for his valuable comments on an earlier version of this manuscript.

## Author Contributions

**Conceptualization:** Annick Lenglet, Omar Contigiani, Cono Ariti, Heiman Wertheim, Joost Hopman.

**Data curation:** Annick Lenglet, Estivern Evens, Kessianne Charles, Carl-Frédéric Casimir, Rodnie Senat Delva, Colette Badjo, Barbara Pawulska.

**Formal analysis:** Annick Lenglet, Omar Contigiani.

**Methodology:** Annick Lenglet.

**Project administration:** Annick Lenglet.

**Resources:** Colette Badjo, Melissa McRae.

**Supervision:** Melissa McRae, Heiman Wertheim, Joost Hopman.

**Validation:** Omar Contigiani, Cono Ariti, Kate Clezy.

**Visualization:** Omar Contigiani.

**Writing – original draft:** Annick Lenglet.

**Writing – review & editing:** Annick Lenglet, Omar Contigiani, Cono Ariti, Estivern Evens, Kessianne Charles, Carl-Frédéric Casimir, Rodnie Senat Delva, Colette Badjo, Harriet Roggeveen, Barbara Pawulska, Kate Clezy, Melissa McRae, Heiman Wertheim, Joost Hopman.

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
