## [Decision Letter · Decision Letter 0]

23 Mar 2022

PONE-D-21-34188Early warning for healthcare acquired infections in neonatal care units in a low-resource setting using routinely collected hospital data: the experience from Haiti, 2014-2018.PLOS ONE

Dear Dr. Lenglet,

Thank you for submitting your manuscript to PLOS ONE. After careful consideration, we feel that it has merit but does not fully meet PLOS ONE’s publication criteria as it currently stands. Therefore, we invite you to submit a revised version of the manuscript that addresses the points raised during the review process.

The manuscript has been evaluated by two reviewers, and their comments are available below.

The reviewers have raised a number of concerns. They request improvements to the reporting of methodological aspects of the study, for example, regarding the definition of sepsis and request more information about patient management. They also request more description of the importance and application of your study.

Could you please carefully revise the manuscript to address all comments raised?

We look forward to receiving your revised manuscript.

Kind regards,

Jamie Royle, PhD

Associate Editor

PLOS ONE

Journal Requirements:

3. In the ethics statement in the manuscript and in the online submission form, please provide additional information about the patient records used in your retrospective study, including: a) whether all data were fully anonymized before you accessed them; b) the date range (month and year) during which patients' medical records were accessed; c) the date range (month and year) during which patients whose medical records were selected for this study sought treatment. If the ethics committee waived the need for informed consent, or patients provided informed written consent to have data from their medical records used in research, please include this information.

This research received no specific grant from any funding agency in the public, commercial or not-for-profit sectors. MSF staff carried out the research as part of their routine roles. The funders had no role in study design, data collection and analysis, decision to publish, or preparation of the manuscript. 

6. We note that you have referenced (ie. Bewick et al. [5]) which has currently not yet been accepted for publication. Please remove this from your References and amend this to state in the body of your manuscript: (ie “Bewick et al. [Unpublished]”) as detailed online in our guide for authors

Reviewers' comments:

Reviewer's Responses to Questions

**Comments to the Author**

1. Is the manuscript technically sound, and do the data support the conclusions?

Reviewer #1: Yes

Reviewer #2: Partly

2. Has the statistical analysis been performed appropriately and rigorously? 

Reviewer #1: Yes

Reviewer #2: Yes

3. Have the authors made all data underlying the findings in their manuscript fully available?

Reviewer #1: Yes

Reviewer #2: No

4. Is the manuscript presented in an intelligible fashion and written in standard English?

Reviewer #1: Yes

Reviewer #2: Yes

5. Review Comments to the Author

Reviewer #1: Comments to the Author:

The significance of conducting the study was well documented. The methodology and data

analysis was sound. The authors also discussed the study limitations.

The recommendations made in the conclusion are reasonable.

Few suggestions for improvement:

Line 64: Add reference for Late-onset sepsis definition that was used

Line 125-127: Please explain further the use of ‘prophylactic’ antibiotics in the neonates with risk factors for sepsis. Was this ‘preemptive therapy’ rather than prophylaxis?

Line 60: No comma required after ‘infection prevention and control’

Line 351-354: Fix the preposition placements in the sentence

Reviewer #2: Thank you for the opportunity to review this manuscript describing an algorithm to predict outbreaks of neonatal sepsis in a low income setting

I have some major concerns with the manuscript in its present form:

• Please confirm your definition of late onset neonatal sepsis. In the introduction you state this is >48 hours, with no reference. Most agree it is > 72 hours

• How was late onset sepsis diagnosed/ defined? Clinically or on the basis of clinical presentation and cultures. Although not explicitly stated, it seems that cultures are expensive and only available in some cases. This definition is crucial to your research question and must be clear.

• Please provide more detail on the management of babies with suspected sepsis – were they all worked up for sepsis – including complete blood count, C reactive protein, cultures etc or were they treated on suspicion and clinical presentation?

• How was antiobiotic usage in your setting determined?

• Please provide more detail on your infection and prevention control management. Did you have a regular review of all patients, cultures and antibiotic usage?

• How did you define an outbreak of sepsis?

• Please clearly distinguish between NCU and NICU in your abstract and methods – in your study setting.

• Please provide more information on the “repeated outbreaks” which is referenced as unpublished data in your introduction. You could add this as an appendix or a data set.

• What is the practical application of your findings? It is self – evident that an increase in cases of late onset sepsis is strongly predictive of an outbreak of late onset sepsis. What does your study add?

6. PLOS authors have the option to publish the peer review history of their article (what does this mean?). If published, this will include your full peer review and any attached files.

Reviewer #1: No

Reviewer #2: **Yes: **Daynia E. Ballot

---

## [Author Response · Author response to Decision Letter 0]

16 May 2022

PONE-D-21-34188

Early warning for healthcare acquired infections in neonatal care units in a low-resource setting using routinely collected hospital data: the experience from Haiti, 2014-2018.

Dear Dr Royle,

Thank you very much for your feedback concerning the above-mentioned manuscript.

We have carefully reviewed each of the reviewers’ comments and thank them for assessing the manuscript. Each comment has been answered or clarified below after ‘Response’.

We have submitted a revised manuscript in marked-up and clean versions as well.

Hopefully, this will address the concerns raised and we look forward to hearing from you.

With best wishes,

Annick Lenglet on behalf of the study team

2. Please include a complete copy of PLOS’ questionnaire on inclusivity in global research in your revised manuscript. 

Response: we have uploaded a completed version of this questionnaire

3. In the ethics statement in the manuscript and in the online submission form, please provide additional information about the patient records used in your retrospective study, including: a) whether all data were fully anonymized before you accessed them; b) the date range (month and year) during which patients' medical records were accessed; c) the date range (month and year) during which patients whose medical records were selected for this study sought treatment. If the ethics committee waived the need for informed consent, or patients provided informed written consent to have data from their medical records used in research, please include this information.

Response: we have included further information in the Ethical Considerations section of the Methods section. It now reads: 

“This study was approved by the National Bioethical Committee of the Ministry of Public Health and Population of Haiti. This research fulfilled the exemption criteria set by the Médecins Sans Frontières Ethics Review Board for a posteriori analyses of routinely collected clinical data and thus did not require MSF ERB review. It was conducted with permission from Melissa McRae, Medical Director, Operational Centre Amsterdam (OCA), Médecins Sans Frontières. 

Patient records included in this retrospective analysis covered the period in which they received treatment from January 2014 to December 2017. They were accessed in February 2022 to start the analysis for this study. Data were aggregated from existing individual health information system records and these aggregated datasets were used for the analysis of this study. The data collected was not collected specifically for the purpose of this study but was part of routinely collected health information systems to monitor programme implementation. Patients therefore did not provide informed consent for the data to be collected.”

This research received no specific grant from any funding agency in the public, commercial or not-for-profit sectors. MSF staff carried out the research as part of their routine roles. The funders had no role in study design, data collection and analysis, decision to publish, or preparation of the manuscript. 

Response: there was no specific funding allocated to this study. All MSF staff carried out the research as part of their routine roles. Thus their salaries were funded by Medecins Sans Frontieres, but not specifically for this study.

Response: the authors designed the study as part of their routine roles in MSF. The organization as such did not influence the study design, data collection, and analysis, decision to publish, or preparation of the manuscript. We have included this statement in the manuscript.

Response: as stated before, all MSF authors participated in this study as part of their routine work for the organization, thus they did receive salaries from MSF even if these were not specifically related to this study. The colleagues from Radboudumc participated in their roles as supervising the first author for her Doctoral studies at this university. Mr Cono Ariti works as a statistician consultant for MSF and consults on analysis plans for all research activities for this Operational Centre of MSF.

Response: this would be the most appropriate statement and was already included in the manuscript

Response: we will also include this in a revised cover letter supporting the revised manuscript, but all the additional information requested is currently reflected in the statement related to Data Availability: “MSF has a managed access system for data sharing that respects MSF’s legal and ethical obligations to its patients to collect, manage and protect their data responsibility. Ethical risks include, but are not limited to, the nature of MSF operations and target populations being such that data collected are often highly sensitive. Data are available on request in accordance with MSF's data sharing policy (available at: http://fieldresearch.msf.org/msf/handle/10144/306501). Requests for access to data should be made to data.sharing@msf.org.” We hope this is sufficient for your purposes.

6. We note that you have referenced (ie. Bewick et al. [5]) which has currently not yet been accepted for publication. Please remove this from your References and amend this to state in the body of your manuscript: (ie “Bewick et al. [Unpublished]”) as detailed online in our guide for authors

Response: I have searched throughout the manuscript and references and am unable to identify any reference related to Bewick et al. The reference number 5 that you refer to in your comment relates to a completely different reference, namely: Gastmeier P, Loui A, Stamm-Balderjahn S, Hansen S, Zuschneid I, Sohr D, et al. Outbreaks in neonatal intensive care units-They are not like others. American Journal of Infection Control. 2007 Apr;35(3):172–6. Is it possible this Editor comment is an error?

Response: we have now included a section after Data availability that is called ‘Supporting information’ with : S1: Outbreaks alarms (red triangles) and indicators for LO sepsis

Comments to the Author

Reviewer #1: Comments to the Author:

The significance of conducting the study was well documented. The methodology and data

analysis was sound. The authors also discussed the study limitations.

The recommendations made in the conclusion are reasonable.

Response: Thank you very much for having reviewed this manuscript, your comments and your recommendations. The manuscript has improved considerably thanks to them. We have tried to take each of your comments on board.

Few suggestions for improvement:

Line 64: Add reference for Late-onset sepsis definition that was used

Response: we did not provide a reference for this as we know that this definition is not always clear and fully agreed upon in the global neonatal health arena. We have now included the following reference: Fanaroff AA, Korones SB, Wright LL, Verter J, Poland RL, Bauer CR, Tyson JE, Philips JB 3rd, Edwards W, Lucey JF, Catz CS, Shankaran S, Oh W. Incidence, presenting features, risk factors and significance of late onset septicemia in very low birth weight infants. The National Institute of Child Health and Human Development Neonatal Research Network. Pediatr Infect Dis J. 1998 Jul;17(7):593-8. doi: 10.1097/00006454-199807000-00004. PMID: 9686724.

Line 125-127: Please explain further the use of ‘prophylactic’ antibiotics in the neonates with risk factors for sepsis. Was this ‘preemptive therapy’ rather than prophylaxis?

Response: pre-emptive therapy refers to prophylactic antibiotics being administered to specific target groups. Indeed, in our context, such pre-emptive treatment was provided to neonates considered at high risk of sepsis. This was based on a clinical judgement using information on prematurity, low birth weight, chorioamnionitis, or other materno-fetal risk factors. 

The sentence has been broken into two sentences and currently reads: “After admission to the NCU following their birth at CRUO, neonates clinically judged to be at risk for sepsis were administered pre-emptive antibiotics. This was based on prematurity, low birth weight, chorioamnionitis, or other materno-fetal risk factors.”

Line 60: No comma required after ‘infection prevention and control’

Response: thank you for your detailed spotting of this mistake. It has now been removed.

Line 351-354: Fix the preposition placements in the sentence

Response: thank you for spotting this strange sentence. We have now split it into two parts and it currently reads like this: “Our findings show that threshold alarms for LO sepsis cases and neonatal mortality performed well to predict HAI outbreaks in an NCU (p<0.0001). This is therefore of particular importance in low-resource settings where microbiological screening or diagnosis of admitted neonates might not be available to detect or confirm outbreaks.”

Reviewer #2: Thank you for the opportunity to review this manuscript describing an algorithm to predict outbreaks of neonatal sepsis in a low income setting

Response: We appreciate the time you took to review the manuscript and your comments to make it a stronger paper.

I have some major concerns with the manuscript in its present form:

• Please confirm your definition of late onset neonatal sepsis. In the introduction you state this is >48 hours, with no reference. Most agree it is > 72 hours

Response: We are aware that there are varying case definitions used for neonatal sepsis in the global arena for neonatal health. There was a recent call to align these definitions published (Molloy, E.J., Wynn, J.L., Bliss, J. et al. Neonatal sepsis: need for consensus definition, collaboration and core outcomes. Pediatr Res 88, 2–4 (2020). https://doi.org/10.1038/s41390-020-0850-5). For the purpose of our hospital and for our study, we have indeed used a neonatal sepsis definition that includes >48hrs. Following feedback from Reviewer 1, we have now inserted a reference for this (Fanaroff AA, Korones SB, Wright LL, Verter J, Poland RL, Bauer CR, Tyson JE, Philips JB 3rd, Edwards W, Lucey JF, Catz CS, Shankaran S, Oh W. Incidence, presenting features, risk factors and significance of late onset septicemia in very low birth weight infants. The National Institute of Child Health and Human Development Neonatal Research Network. Pediatr Infect Dis J. 1998 Jul;17(7):593-8. doi: 10.1097/00006454-199807000-00004. PMID: 9686724). As the patient population in our study was often premature and low birthweight upon admission, we wanted to use a more conservative definition of late onset sepsis. 

• How was late onset sepsis diagnosed/ defined? Clinically or on the basis of clinical presentation and cultures. Although not explicitly stated, it seems that cultures are expensive and only available in some cases. This definition is crucial to your research question and must be clear.

Response: we used a clinical definition for sepsis in neonates since very limited additional laboratory testing was available due to the low resource nature of the hospital. Access to cultures was even further limited. We described the management of this outbreak in a previous paper (Lenglet A, Faniyan O, Hopman J. A Nosocomial Outbreak of Clinical Sepsis in a Neonatal Care Unit (NCU) in Port-Au-Prince Haiti, July 2014 - September 2015. PLoS Curr. 2018 Mar 21;10:ecurrents.outbreaks.58723332ec0de952adefd9a9b6905932. doi: 10.1371/currents.outbreaks.58723332ec0de952adefd9a9b6905932. PMID: 29637010; PMCID: PMC5866103.). There, we also further described the clinical case definition used for sepsis: a neonate in the neonatal unit who presented with one or more of the following clinical signs: prolonged capillary refill, certain skin changes (redness, sclerema), distended abdomen with/without hemorrhagic, brownish or bilious gastric aspirates, tachypnea, tachycardia, persistent jaundice, unstable temperature, signs of disseminated intravascular coagulation (bleeding from catheter sites, bloody secretions from nose and mouth, petechiae), reduced muscle tone, lethargy, apathy or irritability.

We have now included this case definition in the ‘settings’ section of the methods and have referenced the original article. We also added an additional clarification sentence: “The case definition did not include any microbiological criteria due to the difficulty in accessing this diagnostic tool in this setting.”

• Please provide more detail on the management of babies with suspected sepsis – were they all worked up for sepsis – including complete blood count, C reactive protein, cultures etc or were they treated on suspicion and clinical presentation?

Response: this is related to your previous comment. Due to the low resource nature of this hospital, the additional laboratory tests performed were very limited. We would prefer not to include the detailed clinical management of the neonates with sepsis in this paper as the focus is really on the proxy indicators aspect of detecting outbreaks. The previous paper provided additional details about the outbreak management which we feel addresses this question (Lenglet A, Faniyan O, Hopman J. A Nosocomial Outbreak of Clinical Sepsis in a Neonatal Care Unit (NCU) in Port-Au-Prince Haiti, July 2014 - September 2015. PLoS Curr. 2018 Mar 21;10:ecurrents.outbreaks.58723332ec0de952adefd9a9b6905932. doi: 10.1371/currents.outbreaks.58723332ec0de952adefd9a9b6905932. PMID: 29637010; PMCID: PMC5866103.).

• How was antiobiotic usage in your setting determined?

Response: In the part called ‘Data sources’ in the Methods section we specify that antibiotic consumption data was based on “NCU pharmacy consumption data for first (ampicillin and gentamicin), second (ceftazidime and amikacin) and third line (meropenem) antibiotics for neonatal sepsis treatment.” We, therefore, did not have individual antibiotic use records for each neonate. It is one of the aspects we have also highlighted in the discussion section as being a limitation.

• Please provide more detail on your infection and prevention control management. Did you have a regular review of all patients, cultures and antibiotic usage?

Response: we agree that this is a crucial question in relation to the management of the outbreak, this information has been described in detail in the original outbreak paper describing the first year of the outbreak response (Lenglet A, Faniyan O, Hopman J. A Nosocomial Outbreak of Clinical Sepsis in a Neonatal Care Unit (NCU) in Port-Au-Prince Haiti, July 2014 - September 2015. PLoS Curr. 2018 Mar 21;10:ecurrents.outbreaks.58723332ec0de952adefd9a9b6905932. doi: 10.1371/currents.outbreaks.58723332ec0de952adefd9a9b6905932. PMID: 29637010; PMCID: PMC5866103.). We have referenced this paper in the introduction to describe the context and hope this is sufficient supplementary reading for those persons interested in these detailed aspects.

• How did you define an outbreak of sepsis?

Response: in the first part of the Data Analysis section of the methods we address this concern: “We had no pre-existing definition for an outbreak of healthcare-acquired neonatal sepsis in CRUO, and microbiological data was only available for a small proportion of suspected neonatal sepsis patients. We therefore defined an outbreak in the NCU as a moment in time in which we observed statistical abnormalities in any of the routinely collected health indicators. “ We think this definition is clear, but would welcome alternative suggestions to make it better.

 • Please clearly distinguish between NCU and NICU in your abstract and methods – in your study setting.

Response: thank you for pointing this out, we have now included a clarification on why we refer to the neonatal unit in Port au Prince, Haiti as an NCU in the introduction. Basically, it is because the unit did not have any central or umbilical catheters and thus the ‘intensive’ care was reduced. The sentence in the introduction now reads: “Between 2011 and 2018, Médecins Sans Frontières (MSF) managed an emergency obstetric emergency hospital and its associated neonatal care unit (NCU; not relying on central or umbilical catheters) in Port-au-Prince, Haiti.”

• Please provide more information on the “repeated outbreaks” which is referenced as unpublished data in your introduction. You could add this as an appendix or a data set.

Response: in the above-mentioned article (Lenglet et al. 2015) some of the data is shown concerning the ‘repeated outbreaks’ for the period between 2014 and 2015. The other repeated outbreaks are to some extent shown in the Figure 1A and 1B where we show the rates of LO sepsis and weekly cases of these. For this reason, there is no other data to refer to.

• What is the practical application of your findings? It is self – evident that an increase in cases of late onset sepsis is strongly predictive of an outbreak of late onset sepsis. What does your study add?

Response: in most high resources that have sophisticated health information systems which allow clinical staff to monitor clinical, epidemiological and microbiological data on their patients in real time, early warning for suspected outbreaks would probably be based on a clustering of clinical cases and microbiological confirmation that would trigger an automated alarm that would be sent to infection prevention teams in the hospital and alert clinical staff immediately. In low resource settings, where such systems do not exist, where microbiological diagnostics are not accessible, clinical staff have no objective ways of identifying an early warning indicator for suspected outbreaks. This was the starting point for the analysis presented in the manuscript. We have shown that very basic clinical data (which does not require sophisticated electronic health information systems) can reliably trigger alarms for suspected sepsis outbreaks in a neonatal care unit. We have also shown that, when analysed within the unique context of a single neonatal care unit, we can identify a unique threshold of individual cases of late onset sepsis that reliably signal an outbreak for the NEXT epidemiological week. This type of information gives a tested and quantitative tool to clinical and infection prevention staff to intervene early in order to avoid further transmission within the hospital.

The added value of the study is to identify the simple routinely collected data that can be used for the purpose of early warning, to evaluate their performance for reliably providing such an early warning, and to quantify thresholds that can be used in this setting. We feel that the discussion and conclusion of the manuscript clearly communicate this added value in its current format.

---

## [Editor Report · Decision Letter 1]

20 May 2022

Early warning for healthcare acquired infections in neonatal care units in a low-resource setting using routinely collected hospital data: the experience from Haiti, 2014-2018.

PONE-D-21-34188R1

Dear Dr. Lenglet,

We’re pleased to inform you that your manuscript has been judged scientifically suitable for publication and will be formally accepted for publication once it meets all outstanding technical requirements.

Kind regards,

Daynia Elizabeth Ballot, PhD

Guest Editor

PLOS ONE

Additional Editor Comments (optional):

Thank you for addressing the reviewers' comments. The paper is now suitable for publication

For transparency, I served as reviewer 2 for the paper
---

## [Editor Report · Acceptance letter]

14 Jun 2022

PONE-D-21-34188R1 

Early warning for healthcare acquired infections in neonatal care units in a low-resource setting using routinely collected hospital data: the experience from Haiti, 2014-2018. 

Dear Dr. Lenglet:

I'm pleased to inform you that your manuscript has been deemed suitable for publication in PLOS ONE. Congratulations! Your manuscript is now with our production department. 

Kind regards, 

on behalf of

Professor Daynia Elizabeth Ballot 

Guest Editor

PLOS ONE